# *In Vitro* Evaluation of the Potential for Drug Interactions by Salidroside

**DOI:** 10.3390/nu15173723

**Published:** 2023-08-25

**Authors:** Philip G. Kasprzyk, Larry Tremaine, Odette A. Fahmi, Jing-Ke Weng

**Affiliations:** 1DoubleRainbow Biosciences Inc., Lexington, MA 02421, USA; jingke.weng@doublerainbowbio.com; 2Tremaine DMPK Consulting, LLC, Merritt Island, FL 32899, USA; tremainelm@gmail.com; 3DDI-Edge Consulting, LLC, Fort Lauderdale, FL 33308, USA; o.fahmi@pharmadvisors.com; 4Whitehead Institute for Biomedical Research, Cambridge, MA 02142, USA; 5Department of Biology, Massachusetts Institute of Technology, Cambridge, MA 02139, USA

**Keywords:** salidroside, drug-drug interaction, CYP450, MAO-A, MAO-B, OATP

## Abstract

Several studies utilizing *Rhodiola rosea*, which contains a complex mixture of phytochemicals, reported some positive drug-drug interaction (DDI) findings based on in vitro CYP450’s enzyme inhibition, MAO-A and MAO-B inhibition, and preclinical pharmacokinetic studies in either rats or rabbits. However, variation in and multiplicity of constituents present in *Rhodiola* products is a cause for concern for accurately evaluating drug-drug interaction (DDI) risk. In this report, we examined the effects of bioengineered, nature-identical salidroside on the inhibition potential of salidroside on CYP1A2, CYP2B6, CYP2C8, CYP2C9, CYP2C19, CYP2D6, and CYP3A4 utilizing human liver microsomes, the induction potential of salidroside on CYP1A2, CYP2B6 and CYP3A4 in cryopreserved human hepatocytes, the inhibitory potential of salidroside against recombinant human MAO-A and MAO-B, and the OATP human uptake transport inhibitory potential of salidroside using transfected HEK293-OATP1B1 and OATP1B3 cells. The results demonstrate that the bioengineered salidroside at a concentration exceeding the predicted plasma concentrations of <2 µM (based on 60 mg PO) shows no risk for drug-drug interaction due to CYP450, MAO enzymes, or OATP drug transport proteins. Our current studies further support the safe use of salidroside in combination with other drugs cleared by CYP or MAO metabolism or OATP-mediated disposition.

## 1. Introduction

The use of herbal medicines existed for centuries in many developing countries, where a large proportion of the population relies on traditional practitioners and the local availability of medicinal plants in order to meet health care needs [1]. Traditional folk medicine used *Rhodiola rosea* to increase physical endurance, work productivity, longevity, resistance to high altitude sickness, and to treat fatigue, depression, anemia, impotence, gastrointestinal ailments, infections, and nervous system disorders [2].

In the United States, many herbal products are marketed and regulated as dietary supplements, a product category that does not require pre-approval of products based on efficacy, safety, or quality. Unlike regulated pharmaceuticals, safety of an herbal product is based on the extent and experience of human use.

Today, individuals often take multiple medications and over-the-counter products, and thus, the risk of drug interactions must be tested and understood [3]. Nature-derived supplement products are generally a mixture of multiple bioactive compounds, all of which could potentially have drug-drug interactions with other medication. For example, van Diermen et al. [4] used dichloromethane, methanol, and water to extract components of dried and powdered roots of *Rhodiola rosea* and structurally identified twelve components, including salidroside.

Drug-drug interactions are one of the most common causes of adverse drug reactions (ADRs) and the risk of potential DDIs grows proportionally with age and the number of drugs used and may result in the development of ADRs. A retrospective study evaluating the correlation of ADRs and DDI from 2011 to 2020 found that a total of 54% of ADRs were identified as preventable [5], with the severity of ADR significantly correlated with age and the number of suspected drugs [6]. There are several articles indicating the potential risk for metabolic interactions, with particular emphasis and observed frequency of CYP3A4 substrates [7] and MAO [8] families of enzymes.

Previous in vivo preclinical and clinical studies with *Rhodiola rosea* extracts reported various drug-drug interactions, although clinical relevance is yet to be determined. A mild inhibitory effect on CYP2C9 probe substrate activities in rats (34% increase in C_max_ but no change in AUC) [9], rabbits (≈2-fold in AUC) [10], and humans (EXP-3174/losartan ratio of 21%) [11] were reported. An in vivo rat study investigating the effect of single and multiple daily (8 days) doses of salidroside at 30 mg/kg/day on a five-probe drug cocktail exhibited a >20% decrease in AUC_0-∞_ in two out of the five cocktail probe substrates [9]. The AUC_0-∞_ for losartan and midazolam each decreased ~2-fold on day 8. Therefore, *Rhodiola rosea* could potentially affect the intracellular concentrations of drugs metabolized by CYP3A4 or CYP2C9 enzyme, particularly those with a narrow therapeutic index such as phenytoin and warfarin.

Testing for potential drug interactions classically involved in vivo studies in preclinical species (typically rat) or clinical testing with commonly used individual drugs. However, current methods include in vitro studies with human-derived enzymes and drug transporters and clinical studies using index drugs and reference inhibitors based on mechanistic understanding of drug-drug interactions. These preclinical and clinical study designs are summarized in FDA guidances [12,13].

Hellum et al. [14] had shown that after ethanol extraction of six different clones of *Rhodiola rosea*, the five main constituents were salidroside, tyrosol, rosavin, rosarin/rosin, and cinnamic alcohol. In vitro studies with recombinant human CYPs indicated an IC_50_ range of 1.7–3.1 μg/mL. Another reported in vitro study using recombinantly expressed MAO-A and MAO-B showed inhibitory effects by some of the constituents of *Rhodiola rosea* root extracts.

Salidroside may influence the disposition of the drugs that are mainly metabolized by these pathways, but it is unknown if the effects observed with *Rhodiola rosea* extracts are due to salidroside or other components and if effects seen in animal studies will occur in humans. This report examines the testing of salidroside that was produced by our *E. coli* strain with in vitro assays using human-derived or recombinantly synthesized enzyme and drug-transporter systems to determine the effects of only salidroside and not the other components of *Rhodiola rosea* or extracts. We examined the effects of salidroside on: (i) the competitive and time-dependent inhibition potential of salidroside on seven CYP isoform activities in human liver microsomes; (ii) the induction potential of salidroside on CYP1A2, CYP2B6, and CYP3A4 in cryopreserved human hepatocytes; (iii) the inhibitory potential of salidroside against recombinant human MAO-A and MAO-B, and (iv) the OATP human uptake transport inhibitory potential of salidroside using transfected HEK293-OATP1B1 and -OATP1B3 cells. Additionally, the protein binding in human and rat plasma were determined and an estimation of the maximum human plasma concentration (C_max_) of salidroside was utilized to evaluate salidroside in vitro at concentrations of clinical relevance.

## 2. Materials and Methods

### 2.1. Reagents

Salidroside (white crystalline powder; 99.3% pure, CAS #10338-51-9) was supplied by LandKind, Lexington, MA, USA, a subsidiary of DoubleRainbow Biosciences Inc. It is a synthetic biology product from fermentation of a bioengineered *E. coli*. β-naphthoflavone, sulfaphenazole, tranylcypromine, ketoconazole, quercetin, terfenadine, tolbutamine, phenacetin, diclofenac, dextromethorphan, verapamil, omeprazole, hydroxybupropion, dimethyl sulfoxide, 2-mercaptoethanol, leflunomide, K_2_HPO_4_, KH_2_PO_4_, kynuramine, rovustatin calcium, and tolbutamine were purchased from Sigma-Aldrich. Quinidine was purchased from International Laboratory-USA. Ticlopidine was purchased from Fluka. Mephenytoin and tienilic were purchased from GlpBio. Midazolam was purchased from Shanghai Yuansi Standard Science and Technology Co., Ltd. (Shanghai, China). Testosterone was purchased from Adamas Reagent. Bupropion was purchased from Tocris. Midazolam and phenobarbitol were purchased from the National Institutes for Food and Drug Control. Rifampicin was purchased from Macklin. Flumazenil was purchased from Shanghai yuanye Biotechnology Co., Ltd. (Shanghai, China). Acetaminophen was purchased from MedChemExpress. 1′-Hydroxymidazolam and Labetalol were purchased from Shanghai ZZBIO Co., Ltd. (Shanghai, China). Amodiaquine was purchased from Abmole. Furafylline and genfibrozil 1-O-β-glucuronide were purchased from MedChemExpress. Paroxetine was purchased from USP. 1-(20Phenyl-propan-2-piperidinyl) propan (PPP) was purchased from flourochem. K_2_HPO_4_ was purchased from SCR while NADPH was purchased from ACROS. Mixed human liver microsomes were purchased from BioreclamationIVT.

Cryopreserved human hepatocytes from four donors were purchased from BioIVT and Novabiosis and stored in a liquid nitrogen tanker before use. William’s E Medium, DPBS, Glutamax, HEPES buffer solution, fetal bovine serum, HBSS, and penicillin-streptomycin were purchased from Gibco. Percoll was purchased from GE Healthcare. Human recombinant insulin was purchased from YEASEN. Dexamethasone was purchased from Alfa Aesar. Plating medium and incubation medium were purchased from BioIVT. Matrigel was from Corning. LDH Cytotoxicity Assay Kit was purchased from Beyotime Biotechnology. High capacity cDNA Reverse Transcription Kit with RNAse Inhibitor, TaqMan Universal PCR Master Mix, Eukaryotic 18S rRNA Endogenous Control, Human CYP3A4 20X Gene Expression Assay labeled with FAM/MGB, Human CYP1A2 20X Gene Expression Assay labeled with FAM/MGB, and Human CYP2B6 20X Gene Expression Assay labeled with FAM/MGB were purchased from Applied Biosystems. RT-PCR grade water was purchased from Invitrogen. Rneasy 96 Kit was from Qiagen. The 96-well cell culture plates and collagen I 48-well clear multiwell plates and human recombinant MAO-A and MAO-B supersomes were purchased from Corning. Salidroside was provided by Double Rainbow Biosciences. The transporter cells and relative cell culture materials HEK-OATP1B1 and HEK-OATP1B3 were purchased from GenoMembrane, while DMEM, DPBS, fetal bovine serum, penicillin-streptomycin solution, non-essential amino acid, and 2-[4-(2-Hydroxyethyl)-1-piperazinyl] ethanesulfonic acid (HEPES) were purchased from Gibco. Estradiol-17β-glucuronide was from Toronto Research Chemicals and cylcosporin A from J & K.

### 2.2. CYP450’s Inhibition Assays

The inhibition potential of salidroside (0.0488–50 µM) on the activities of CYP1A2, CYP2B6, CYP2C8, CYP2C9, CYP2C19, CYP2D6, and CYP3A4 in human liver microsomes were tested. Prototypical positive controls (β-naphthoflavone (0.0039–0.4 µM), ticlopidine (0.0012–1.2 µM), quercetin (0.0039–4 µM), sulfaphenazole (0.0012–1.2 µM), tranylcypromine (0.0117–12 µM), quinidine (0.0001–0.12 µM), and ketoconazole (0.0004–0.40 µM)) were included.

Test compound/controls working solution or DMSO were added in 238.5 µL of liver microsomes working solution, mixed well by pipetting several times and pre-incubated in a 37 °C shaking water bath for 5 min. Following the pre-incubation, 60 µL of NADPH working solution were added, mixed well by pipetting several times, and incubated in a 37 °C shaking water bath for 10 min. Following the incubation, 300 µL of quenching solution were immediately added and vortexed for ~20 s to mix. All samples were centrifuged at 4000 rpm for 15 min at 4 °C and then the supernatant was transferred for LC-MS/MS analysis. IC_50_ values were determined by the Hill slope equation using Graph Pad Prism.

The potential for time dependent inhibition (TDI) of salidroside was tested by the IC50 shift assay for CYP1A2, CYP2B6, CYP2C8, CYP2C9, CYP2C19, CYP2D6, and CYP3A4, utilizing a pool of human liver microsomes (Lot ZZQ). Typical substrates (phenacetin (50 μM), bupropion (80 μM), amodiaquine (2 μM), diclofenac (10 μM), S-mephenytoin (25 μM), dextromethorphan (5 μM), and testosterone (50 μM), respectively). A specific amount of each positive control compound (100 mM furafylline, 30 mM tienilic acid, 20 mM ticlopidine, 20 mM paroxetine, 100/20 mM verapamil, 50 mM 1-(2-phenylpropan-2-piperidinyl) propan (PPP), and 20 mM gemfibrozil 1-O-β-glucouronide) was independently prepared as stock solutions in DMSO, then the first five stock solutions were equally mixed into a five-in-one stock solution. Incubation without 30 min pre-incubation, 178.5 µL of liver microsomes working was pre-incubated in a 37 °C water bath for 30 min. After the pre-incubation, 1.5 μL of test compound or control compound working solution, 60 μL of substrate working solution, and 60 μL of NADPH working solution were added; then, the reaction mixture was mixed by pipetting up and down and then incubated at 37 °C for 10 min. Incubation with 30 min pre-incubation. Next, 1.5 μL of test compound or control compound working solutions and 60 μL of NADPH working solution were added into 178.5 μL of liver microsome working solution, then the reaction mixture was mixed by pipetting up and down and then pre-incubated at 37 °C for 30 min. After the pre-incubation, 60 μL of substrate working solution was added, then the reaction mixture was mixed by pipetting up and down and then incubated at 37 °C for 10 min. Following the incubation, 300 μL of quenching solution was added into the corresponding wells to terminate the reaction. All the above samples were vortexed for 1 min and centrifuged at 4000 rpm, 4 °C for 15 min. After centrifugation, the supernatant was removed for further LC-MS/MS analysis. IC_50_ shift is determined by using the equation of IC_50 without 30 min preincubation_/IC_50 with 30 min preincubation._ IC_50_ shift value of >1.5-fold is considered positive for TDI.

### 2.3. CYP450’s Induction Assays

The induction potential of salidroside on CYP1A2, CYP2B6, and CYP3A4 in cryopreserved human hepatocytes was tested. The four donors’ information is listed in Table 1 below:

All hepatocyte incubations were conducted at 37 °C, 95% air/5% CO_2_. The sandwich medium was removed, and the hepatocytes were treated with incubation solutions containing salidroside (1, 5, 10, and 30 µM), vehicle (0.1% DMSO), or positive inducers in triplicate for 24 h after the cultures were established. The incubation solution was aspirated and replaced with incubation solution containing the same concentrations of salidroside, vehicle, or positive inducers for an additional 24 h. The total treatment period was 48 h. After the treatment period of 48 h, the incubation solution was aspirated. Hepatocytes were washed twice with pre-warmed HBSS containing 10 mM HEPES (HBSS I) and then the remaining solution was aspirated. Hepatocytes were washed once with pre-warmed HBSS I and incubated with 140 μL of RLT (lysis buffer from RNeasy 96 Kit) supplemented with 1% β-mercaptoethanol. Plates were then stored at −70 °C in a freezer until RNA analysis. Before RNA extraction, the frozen plate was thawed, and cell lysates were transferred to 96-well plates. RNA isolation was performed using a QIAcube RNA extraction system. Reverse transcription was performed to obtain cDNA. Quantification of the selective genes by real-time quantitative polymerase chain reaction (qPCR) was performed with TaqMan universal PCR Master Mix on the Bio-rad CFX384 TouchTM Real-Time PCR System. The relative change in the mRNA level of the selected target genes induced by each test compound was expressed in relation to the vehicle control sample, and fold changes in gene expression were determined by the ΔΔCt method.

### 2.4. Evaluation of the Inhibitory Potential of Salidroside against Human MAO-A and MAO-B

A total of 99 µL of substrate working solution was added into 1.1 mL incubation tubes and then 1 µL of the test compound/control working solution or DMSO (vehicle control) was added, mixed well, and pre-incubated at 37 °C for 5 min. MAO-A (0.005 mg/mL) and MAO-B (0.02 mg/mL) were pre-incubated in the working solution at 37 °C for 5 min. A total of 100 µL of pre-incubated MAO-A or MAO-B working solution was added to incubation tubes to start the reaction, mixed well, and incubated at 37 °C for 15 min. Following the incubation, immediately add 200 µL of quenching solution were immediately added and vortexed to mix. All samples were centrifuged at 4000 rpm at 4 °C for 15 min, then 100 µL supernatant with 100 µL of water was transferred for LC-MS/MS analysis. IC_50_ values were determined by the Hill slope equation using Graph Pad Prism.

### 2.5. Assessment of the Transport Potential of Salidroside Using HEK293-OATP1B1 and -OATP1B3 Cells

HEK293-OATP1B1 and OATP1B3 cells were diluted to 1 × 10^6^ cells/mL with DMEM supplemented with 1% non-essential amino acid (NEAA), 10% fetal bovine serum (FBS), 1% PS solution (100 U/mL penicillin-G and 100 μg/mL streptomycin), and seeded on a 96-well BioCoat™ poly-d-lysine plate at a density of 1 × 10^6^ cells/mL. The cells were incubated at 37 °C, 5% CO_2_/95% air and saturated humidity for 3–4 h. Once the cells were anchored on the plate, the cell culture was replaced with DMEM containing 10% FBS and 1% PS solution (100 U/mL penicillin-G and 100 μg/mL streptomycin solution) and cultured one day for the uptake and inhibition studies. The cell culture medium was removed from the 96-well plate, then rinsed twice with 0.1 mL/well of pre-warmed transport buffer. The transporter cells containing pre-incubation solution were pre-incubated at 37 °C for 30 min. Then the pre-incubation solution was removed and replaced with 0.1 mL/well of dosing solution containing a substrate, respectively (blank transporter buffer containing a substrate, transport buffer containing a substrate and PC, and transport buffer containing a substrate and salidroside). Then, the transporter cells were incubated at 37 °C for 2 min. After incubation, all solutions in the cells were discarded and rinsed with 0.10 mL/well of ice-cold transport buffer (4 °C) three times. After rinsing, 0.10 mL/well of distilled water was added, and the transporter cells were lysed via three cycles of freezing and thawing of liquid N_2_ (−196–37 °C). A total of 50 μL of cell lysate was quenched with methanol and acetonitrile containing 100 ng/mL tolbutamide and 100 ng/mL labetalol (as internal standards) at a volume ratio of 1:2. All samples were shaken and centrifuged at 4000 rpm at 4 °C for 15 min and the supernatant was removed for future analysis by LC-MS/MS.

### 2.6. Plasma Protein Binding

DMSO solutions of salidroside and warfarin (positive control) were spiked into human and SD rat plasma to 5 μM concentrations (0.5% final DMSO concentration). The experiment was performed using equilibrium dialysis with the two compartments separated by a semi-permeable membrane. Phosphate-buffered saline (PBS) (pH 7.4) was added to one side of the membrane and the plasma solution to the other side. After equilibration (5 h at 37 °C), samples were taken from both sides of the membrane, transferred to a new plate, and were matrix-matched by the addition of an appropriate amount of PBS to the plasma samples and blank plasma to the PBS samples. Methanol/acetonitrile (1:1/vv, three volumes) containing an analytical internal standard terfenadine or tolbutamide (IS) was added to precipitate the proteins and extract salidroside and IS. Incubations were performed in duplicate. Warfarin served as a control. The fraction unbound in plasma (fu,p) = Cb/Cp, where Cb represents the concentration in buffer and Cp the concentration in plasma at 5 h.

### 2.7. Bioanalytical Methods

The following compounds were quantitatively determined by LC-MS/MS methods.

For the transporter-transfected HEK293 cells transport inhibition study, the concentrations of rosuvastatin and estradiol-17β-glucuronide were used, with tolbutamide as the internal standard and for CYP induction by enzyme activity, the metabolites acetaminophen, hydroxybupropion, and 1′-hydroxymidazolam. For MAO-A and MAO-B inhibition, 4-hydroxyquinoline with tolbutamide was used as the internal standard. Salidroside and warfarin were measured in the plasma protein binding study. The HPLC-MS system utilized a Shimadzu LC 30 series instrument and Sciex QTRAP 5500 mass spectrometer 5500 with turbo ionspray (ESI in positive mode for all analytes except salidroside, which was run in the negative mode). Samples were injected onto a Kinetex 2.6 µm C18 100A column (3.0 mm × 50 mm) and quantified by reaction monitoring of metabolites and internal standard. The transporter study employed a calibration curve of spiked samples and inclusion QC samples to confirm assay performance. The MAO inhibition and plasma protein binding studies used peak area ratios (PAR) of analyte/internal standard to semi-quantitatively determine the concentration of 4-hydroxyquinoline by the LC-MS/MS method or the relative concentrations of salidroside and warfarin in the buffer and plasma aliquots.
Enzyme activity (%) = PAR with compound/PAR vehicle × 100.

The vehicle sample was the metabolite peak area ratio in the incubation lacking salidroside as an inhibitor.

For competitive and time-dependent CYP inhibition, the peak area ratios (PAR) of analyte/internal standard were used to semi-quantitatively determine the concentration of acetaminophen (CYP1A2 activity), 4’-hydroxydiclofenac (CYP2C9), 4’-hydroxymephenytoin (CYP2C19), dextrorphan (CYP2D6), 1’-hydroxymidazolam (CYP3A4), 6β-hydroxytestosterone (CYP3A4), hydroxybupropion (CYP2B6), and N-desethylamodiaquine (CYP2C8), with terfenadine as the internal standard by LC-MS/MS method. The HPLC-MS system utilized an Acquity UPLC system and Sciex API 4000 mass spectrometer in ESI positive mode, with quantitation employing MRM. Samples were injected onto an Atlantis T3 5 μM column (2.1 x 50 mm).

## 3. Results

### 3.1. Evaluation of the Potential for the Direct Inhibition of Salidroside on Human Cytochrome P450 Enzymes

The potential for CYP inhibition of salidroside towards cytochrome P450 (CYP) enzymes was evaluated using a mixed donor pool of human liver microsomes. The probe substrates phenacetin (CYP1A2), bupropion (CYP2B6), amodiaquine (CYP2C8), diclofenac (CYP2C9), (*S*)-mephenytoin (CYP2C19), dextromethorphan (CYP2D6), testosterone (CYP3A4/5), and midazolam (CYP3A4/5) were used as markers for CYP activities in human liver microsomes. The IC_50_ values for the positive controls were in the range for historical data (Table 2).

When salidroside was incubated at several concentrations (0.5–50 µM), there was no measurable inhibition in marker substrate metabolic rate for any of the seven CYP isoforms tested. The IC_50_ values of salidroside for the direct inhibition of CYP1A2, CYP2B6, CYP2C8, CYP2C9, CYP2C19, CYP2D6, and CYP3A4/5 (as measured by testosterone 6β-hydroxylation and midazolam 1′-hydroxylation) were all >50 µM, as summarized in Table 2.

### 3.2. Evaluation of the Potential for Time-Dependent Inhibition of Salidroside on Human Cytochrome P450 Enzymes

The potential for time-dependent inhibition (TDI) of salidroside towards cytochrome P450 (CYP) enzymes was also evaluated using the same lot of human liver microsomes. The experiments measured the metabolic rates of marker substrates with or without a 30 min pre-incubation of salidroside (up to 50 µM) or positive controls (known inactivators) in the presence of NADPH. The probe substrates phenacetin (CYP1A2), bupropion (CYP2B6), amodiaquine (CYP2C8), diclofenac (CYP2C9), (*S*)-mephenytoin (CYP2C19), dextromethorphan (CYP2D6), testosterone (CYP3A4/5), and midazolam (CYP3A4/5) were used as markers for CYP activities in human liver microsomes. Positive control inhibitors for the time-dependent inhibition were included within each assay and showed a significant shift in the IC_50_ values, which is consistent with the literature (Table 3).

Salidroside did not appear to cause time-dependent inhibition of any CYP enzyme activity examined since there was no shift in IC_50_ values observed after salidroside (up to 50 µM) was preincubated with human liver microsomes in either the presence or absence of NADPH for 30 min (Table 3).

### 3.3. Evaluation of the Potential of Salidroside on Induction of Human Cytochrome P450 Enzymes

The potential for salidroside to induce CYP1A2, CYP2B6, and CYP3A4 in cryopreserved human hepatocytes was initially evaluated in a single donor (Lot# SNP—hepatocyte cell viability—90.1%) at 0.1, 1.0, and 10 µM by enzyme activity. Based on the results of the preliminary study, a full study in three different lots of human hepatocytes (Lots# RFM—hepatocyte cell viability—87.2%, # 1911442 hepatocyte cell viability—91.4%, and # MWB hepatocyte cell viability—91.0%) at concentrations up to 30 µM was evaluated based on mRNA data only.

### 3.4. Effect of Salidroside on Enzyme Activity

CYP1A2, CYP2B6, and CYP3A4 activities in cryopreserved human hepatocytes were quantified by measuring the formation of acetaminophen from phenacetin, hydroxybupropion from bupropion, and 1-hydroxymidazolam from midazolam. Following treatment of cryopreserved human hepatocytes prepared from donor (SNP) with the prototypical inducers 50 μM omeprazole for CYP1A2, 1000 μM phenobarbital for CYP2B6, and 10 μM rifampicin for CYP3A4/5, activities were increased 3.55-, 5.69-, and 2.79-fold of vehicle control, respectively. CYP1A2, CYP2B6, and CYP3A4 activity levels in hepatocytes treated with salidroside at the concentrations of 0.1, 1, and10 µM were not increased over the positive controls in donor (SNP) (Table 4), indicating that salidroside at concentrations up to 10 μM had no demonstrable induction to CYP1A2, CYP2B6, and CYP3A4 activities.

### 3.5. Effect of Salidroside on CYP3A4 mRNA Expression

Induction of CYP1A2, CYP2B6, and CYP3A4 was observed for the known inducers rifampin, phenobarbital, and omeprazole as measured by mRNA levels. Treatment of the human hepatocytes with salidroside did not cause induction of CYP1A2, CYP2B6, and CYP3A4 mRNA levels, as summarized in Table 5.

### 3.6. Evaluation of the Inhibitory Potential of Salidroside against Human MAO-A and MAO-B

The oxidation of kynuramine to 4-hydroxyquinoline generated was the in vitro marker reaction of MAO-A and MAO-B enzymes. In this study, the MAO-A and MAO-B enzyme activities were measured and the IC_50_ values of the known inhibitor leflunomide were 13.8 μM and 7.40 μM (Table 6 and Figure 1), respectively, which were within the acceptable range of historical data.

The IC_50_ values of salidroside (0.5–50 µM) towards MAO-A and MAO-B were greater than 50.0 µM, with a maximum % inhibitions at 50 µM of 13.3% and 22.5%, respectively (Table 6 and Figure 1), indicating that salidroside showed minimal inhibition activity of MAO-A and MAO-B enzymes.

### 3.7. Evaluation of the Transport Inhibition Potential of Salidroside to Hepatic Uptake Drug Transporters OATP1B1 and OATP1B3

Estradiol-17β-glucuronide (substrate) was incubated with an OATP1B1-transfected HEK293 cell monolayer in the presence or absence of salidroside at concentrations up to 100 µM. IC_50_ values for the positive control inhibitor of OATP1B1 (cyclosporin A) was 0.0714 µM (Figure 2).

The mean % of control activity in the presence of salidroside was unchanged up to concentrations of 33.3 mM and only reduced to 84.4% at 100 µM and the calculated IC_50_ was >100 µM (Figure 2).

Rosuvastatin (substrate) was incubated with an OATP1B3-transfected HEK293 cell monolayer in the presence or absence of salidroside at concentrations up to 100 µM or the positive control rifampicin. The calculated IC_50_ value for the positive control inhibitor of OATP1B3 (rifampicin) was 1.50 µM (Figure 3). The mean control activities (% of control activity) in the presence of salidroside were unchanged at concentrations up to 100 mM, and therefore, the calculated IC_50_ was estimated to be >100 µM.

### 3.8. Protein Binding of Salidroside in Human and Rat Plasma

The protein binding of salidroside was determined in human and plasma by equilibrium dialysis at a concentration of 5.0 µM. Salidroside was minimally bound in plasma, as the unbound fractions in plasma (f_u,plasma_) were 0.985 and 0.856 in human and rat, respectively.

### 3.9. Estimation of Salidroside Maximum Plasma Concentration

An estimation of the maximum plasma concentration of salidroside in humans was made from the pharmacokinetics in rats after oral dosing [9] and the recommended human daily dose of salidroside of 60 mg (two 30 mg capsules). Male Sprague–Dawley rats were dosed either with IV or orally at 12 mg/kg (N = 6/group). Salidroside was moderately cleared (21 mL/min/kg) and well absorbed orally. With an oral bioavailability (f_b_) of 32.1%, the estimated oral absorption (f_a_) would be f_a_ = f_b_/((1 − CL/Q_h_) × 100%) or 46%. After oral administration, the observed C_max_ was 4.3 mg/mL or 14.3 µM. Based on the 12-fold higher oral dose in rat relative to humans (60 mg daily or ~1 mg/kg), the equivalent predicted human total C_max_ is ~1.2 µM and the unbound C_max_ is ~1.18 μM (C_max_ x f_u,plasma_).

## 4. Discussion

People taking extracts of *Rhodiola rosea* reported a variety of beneficial effects [1]. Marketed and regulated as a dietary supplement, it is a product category that does not require regulatory pre-approval based on efficacy, safety, or quality. Moreover, marketed extracts contain a mixture of multiple bioactive compounds, varying in composition by different manufacturers. Previous in vitro and in vivo preclinical and clinical studies with *Rhodiola rosea* extracts reported various drug-drug interactions, although causative agent(s) and clinical relevance are yet to be determined. To determine if the effects observed with Rhodiola rosea extracts are specifically due to Salidroside, we tested a biosynthesized form currently available commercially. Using experimental designs recommended by the FDA, bioengineered salidroside of high purity was without either inhibitory or inductive effects on CYPs or inhibitory effects on MAO-A, MAO-B enzymes or OATP1B1 or OATP1B3 drug transporters.

Cytochrome P450 (CYP) metabolism is a common pathway for the modification and breakdown of marketed pharmaceuticals in the human body. Seven CYP isoforms (CYP3A4, CYP2D6, CYP2C9, CYP2C19, CYP2C8, CYP2B6, and CYP1A2) account for >90% of enzymatic reactions catalyzed by P450s. Regarding drug metabolism, three-quarters of the human P450 reactions can be accounted for by a set of five P450s: 1A2, 2C9, 2C19, 2D6, and 3A4, and the largest fraction of the P450 reactions is catalyzed by P450 3A [15]. The cytochrome P450s are responsible for about 75% of the phase I-dependent drug metabolism and for the metabolism of a huge amount of dietary constituents and endogenous chemicals [16].

Salidroside was not previously studied regarding potential CYP inhibition or induction using human-derived reagents. Induction of CYP1A2, CYP2B6, CYP3A4, and CYP2C9 was inferred based on the pharmacokinetic alteration of drugs in male SD rats, using specific drugs that are selective substrates for these human CYP isoforms. Three groups of six animals were used, the control receiving no pretreatment, the second receiving 30 mg/kg salidroside daily for 8 days, and the third group received 30 mg/kg salidroside on day 8. All three groups were administered the cocktail of five drugs on day 8; 2 h after salidroside or placebo administration. Reductions in AUC and t_1/2_ were observed for phenacetin, bupropion, losartan, and midazolam in the single-dose salidroside-pretreated group. While the authors concluded this reflected CYP induction [9], this group should have been evaluating potential inhibition, since induction via the respective nuclear receptors AhR, CAR, and PXR for new protein synthesis does not occur that rapidly. The AUC_0-∞_ for losartan and midazolam each decreased ~2-fold following eight daily doses of salidroside. Therefore, *Rhodiola rosea* could potentially affect the intracellular concentrations of drugs metabolized by CYP3A4 or CYP2C9 enzyme. However, there are important species differences in the ligand preferences of these xenobiotic-sensing receptors limiting the translation of in vivo results from the experimental animals to the humans [17].

In this study, both the potential competitive and time-dependent CYP inhibition and CYP induction by salidroside were assessed over a concentration range using in vitro assays employing human-derived reagents, as recommended by the FDA [12]. At concentrations up to 50 μM, there was no inhibition of CYP1A2, CYP2B6, CYP2C8, CYP2C9, CYP2C19, CYP2D6, or CYP3A4. To assess CYP induction, salidroside was incubated in three lots of cryopreserved human hepatocytes for 48 h at concentrations up to 30 μM, and CYP mRNA was measured. There was no induction of CYP3A4, CYP1A2, or CYP2B6 mRNA.

Monoamine oxidases (MAOs) regulate the metabolic degradation of catecholamines and serotonin in the central nervous system and peripheral tissues and inhibition can alter the breakdown of these important biomolecules. Toxicity resulting from MAO inhibition includes serotonin syndrome and hypertension [18,19]. Interactions due to inhibition of monoamine oxidase A (MAO-A) or MAO-B were extensively studied due to the historical use of MAO inhibitors as therapeutic agents, originally in the treatment of depression and later in the management of affective and neurological disorders, stroke, and aging-related neurocognitive changes [19]. Despite the efficacy in various atypical and treatment-resistant depression, MAO inhibitors remain underutilized due to significant associated adverse effects, specifically hypertensive crisis and serotonin syndrome. This concern is further exacerbated due to the robust list of drugs and foods that must be avoided or consumed with caution in order to minimize the potential for adverse reactions [18,19].

Extraction mixtures of *Rhodiola rosea* and twelve individual, isolated components were previously tested for MAO-A and MAO-B inhibitory potency [4]. At a concentration of 100 µg/mL crude mixtures, dichloromethane, methanol, and water extracts were found to have 50.5–92.5% inhibition of MAO-A and 66.9–88.9% inhibition of MAO-B. Of the twelve isolated compounds, each tested at a single concentration of 10 µM, no individual compound had similar inhibitory potency for MAO-A, while rosirdin had an 83.8% inhibition of MAO-B. Salidroside was inactive towards MAO-A and had 35.8% inhibition of MAO-B. Our findings indicate no inhibition of MAO-A by salidroside and found only a 22.5% inhibition of MAO-B at 50 µM.

Recent reports suggest that the rate-limiting step in the overall clearance of several major classes of therapeutics, including statins and sartans, is hepatic uptake by OATP1B1 and/or 1B3 [20]. The potential drug interaction of *Rhodiola rosea* extract at 50 mg/kg in female New Zealand rabbits on the pharmacokinetics of losartan (5 mg/kg orally), a drug whose clearance is mediated in part by OATP hepatic uptake and by CYP2C9 and CYP3A4 metabolism, was studied. The AUC and t_1/2_ of losartan were increased ~2-fold [10]. Therefore, our studies were conducted to specifically evaluate the active component salidroside at concentrations up to 100 µM, on the uptake of marker substrates by HEK293 cells individually transfected with OATP1B1 or OATP1B3. There was no inhibition of OATP1B1 or OATP1B3 at concentrations of 33.3 or 100 µM, respectively, and a modest 15.5% inhibition of OATP1B1 at 100 µM.

The potential for DDI interactions altering the levels of salidroside were not evaluated due to the balanced clearance of metabolism and renal clearance of salidroside, such that a complete inhibition of either mechanism would only result in a maximum 2-fold increase. Two studies determined the PK, tissue distribution, and excretion of salidroside following IV doses in rats. The percentage recovery of an IV dose as unchanged in the urine was 53.7 and 64% [3,21]. Three metabolites of salidroside were identified in rat urine after oral administration; p-Tyrosol, formed by deglycosylation and a glucuronide or sulfate conjugate to the phenolic oxygen of salidroside [22].

Thus, because of these results, engineered salidroside would provide users a better product since one of the main ingredients touted in *Rhodiola rosea* extracts for activity is salidroside [23], which allows no intereference from other compounds in the extract mixture. Additionally, since there is limited availability of the plants, challenging long maturation time from planting to harvest (~5 years), and labor-intensive harvesting and processing, there are no supply chain issues [24].

## 5. Conclusions

We previously reported that the bioengineered salidroside is safe and not genotoxic. In addition, the no observed adverse effect level (NOAEL) for the bioengineered salidroside is ≥2000 mg/kg/day based on the multiple oral dose rat study [25].

These prior safety results underscore the safety of a bioengineered salidroside product, even at high doses. In this paper, the results demonstrate that the in vitro testing of salidroside at concentrations exceeding the predicted plasma concentrations of <2 µM (based on 60 mg PO) showed no risk for drug-drug interaction due to CYP450, MAO enzymes, or OATP drug transport proteins. Earlier studies with positive findings appear to be caused by other constituents of *Rhodiola rosea* extracts or conclusions based on the in vivo effects in preclinical species. Together, the results from experimental designs recommended by the FDA indicate that bioengineered salidroside of high purity can be safely applied as an ingredient in dietary supplements, as a food ingredient, and potentially as a pharmaceutical therapeutic agent, without risk of drug interaction.

## Figures and Tables

**Figure 1 nutrients-15-03723-f001:**
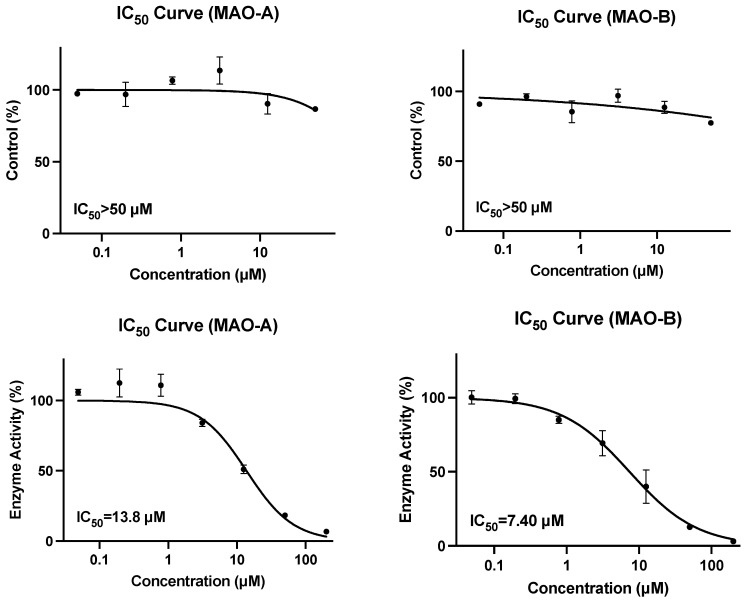
Inhibition curve of salidroside and positive control to MAO-A and MAO-B; upper plots are for salidroside, and bottom plots are for the positive control leflunomide.

**Figure 2 nutrients-15-03723-f002:**
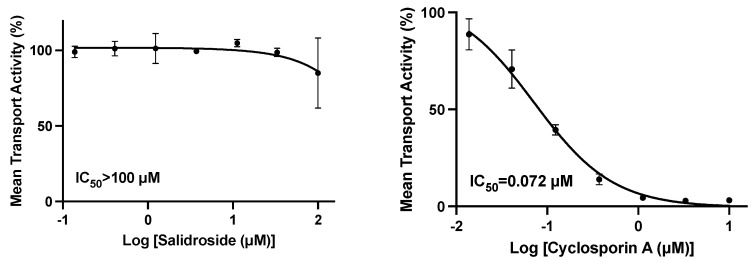
Effect of salidroside and the positive control cyclosporin A on the uptake of estradiol-17β-glucuronide in HEK293-OATP1B1 cells; the left plot is salidroside and the right plot is cyclosporin A.

**Figure 3 nutrients-15-03723-f003:**
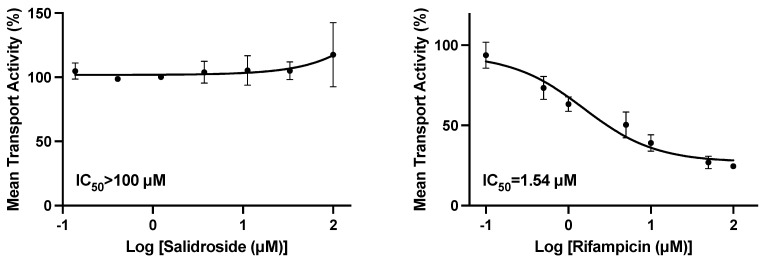
Effect of salidroside on the uptake rate of rosuvastatin in HEK293-OATP1B3 cells; the left plot is salidroside and the right plot is rifampicin.

**Table 1 nutrients-15-03723-t001:** Donor characteristics.

Donor	Lot Number	Gender	Age (Years)	Ethnicity	Tobacco Use	Cause of Death
1	SNP	Male	62	American Indian	1 ppd × 20 years	N/A
2	RFM	Female	73	Caucasian	Smoked ½ ppd since adulthood	N/A
3	199442-01	Female	45	Caucasian	Unknown	N/A
4	MWB	Male	25	Caucasian	Yes	N/A

**Table 2 nutrients-15-03723-t002:** CYP Inhibition (IC_50_ in µM *) results of salidroside (0.05–50 µM) and the positive controls.

	Salidroside (IC_50_)	Positive Controls: IC_50_ Values
CYP1A2	>50	Naphthoflavone: 0.144
CYP2B6	>50	Ticlopidine: 0.0617
CYP2C8	>50	Quercetin: 3.74
CYP2C9	>50	Sulfaphenazole: 0.524
CYP2C19	>50	Tranylcypromine: 9.12
CYP2D6	>50	Quinidine: 0.0451
CYP3A4/5 (Midazolam)	>50	Ketoconazole: 0.0270
CYP3A4/5 (Testosterone)	>50	Ketoconazole: 0.0183

* Mean of N = 2 replicates.

**Table 3 nutrients-15-03723-t003:** IC_50_ * shift determinations for salidroside and the positive controls.

	Salidroside			Positive Controls		
CYP450	IC50 (μM) after 30 min Preincubation without NADPH	IC50 (μM) after 30 min Preincubation with NADPH	IC50 Shift	IC50 (μM) after 30 min Preincubation without NADPH	IC50 (μM) after 30 min Preincubation with NADPH	IC50 Shift
CYP1A2	>50	>50	NA	4.50	0.300	15
CYP2B6	>50	>50	NA	2.96	0.284	10.4
CYP2C8	>50	>50	NA	34.4	1.59	21.6
CYP2C9	>50	>50	NA	0.567	0.0654	837
CYP2C19	>50	>50	NA	0.435	0.128	3.4
CYP2D6	>50	>50	NA	0.253	0.0342	7.4
CYP3A4/5 (M)	>50	>50	NA	25.0	2.20	11.4
CYP3A4/5 (T)	>50	>50	NA	13.0	1.40	9.3

* Mean of N = 2 replicates.

**Table 4 nutrients-15-03723-t004:** CYP1A2, CYP2B6, and CYP3A4/5 activity in cryopreserved hepatocytes (Lot# SNP).

CYP450	Sample ID	Mean Enzyme Activity (pmol/10^6^ cells/min) *	Mean Fold Induction	SD	% of Positive Control
CYP1A2	VC (0.1% DMSO)	2.73	1.00	0.22	
	OME-50 µM	9.68	3.55	0.20	100
	Salidroside-0.1 µM	2.18	0.80	0.25	−8
	Salidroside-1 µM	1.81	0.66	0.27	−13
	Salidroside-10 µM	2.08	0.76	0.06	−9
CYP2B6	VC (0.1% DMSO)	0.88	1.00	0.17	
	PB-1000 µM	5.01	5.69	0.71	100
	Salidroside-0.1 µM	0.77	0.87	0.24	−3
	Salidroside-1 µM	0.51	0.58	0.34	−9
	Salidroside-10 µM	0.76	0.87	0.23	−3
CYP3A4	VC (0.1% DMSO)	2.51	2.85	0.38	
	RIF-10 µM	6.98	7.94	0.24	100
	Salidroside-0.1 µM	1.82	2.07	0.44	−15
	Salidroside-1 µM	1.36	1.55	0.65	−26
	Salidroside-10 µM	1.88	2.13	0.20	−14

* Mean of N = 3 replicates.

**Table 5 nutrients-15-03723-t005:** CYP1A2, CYP2B6, and CYP3A4 gene expression in cryopreserved human hepatocytes from three individual donors.

		Lot# RFM			Lot# 1911442			Lot# MWB		
	Compound ID	Mean Fold Induction *	SD	% Positive Control	Mean Fold Induction *	SD	% Positive Control	Mean Fold Induction *	SD	% Positive Control
CYP1A2	VC (0.1% DMSO)	1.0	0.2		1.0	0.3		1.0	0.3	
	Omeprazole/50 μM	19	1.6	100%	10	0.9	100%	24	6.0	100%
	Salidroside/1 μM	0.9	0.1	−1%	0.9	0.2	−1%	1.3	0.2	1%
	Salidroside/3 μM	0.9	0.2	0%	1.0	0.1	0%	0.8	0.1	−1%
	Salidroside/10 μM	0.9	0.1	0%	1.1	0.6	2%	0.8	0.3	−1%
	Salidroside/30 μM	0.8	0.1	−1%	0.9	0.2	−1%	1.2	0.3	1%
CYP2B6	VC (0.1% DMSO)	1.0	0.1		1.2	0.9		1.0	0.3	
	PB 1000 µM	8.81	1.0	100%	13.6	14	100%	14.7	0.9	100%
	Salidroside/1 μM	0.9	0.2	−1%	0.6	0.3	−3%	1.2	0.0	1%
	Salidroside/3 μM	1.2	0.6	3%	0.9	0.1	−1%	0.9	0.3	0%
	Salidroside/10 μM	0.8	0.2	−3%	1.1	0.2	0%	1.2	0.1	2%
	Salidroside/30 μM	0.6	0.1	−5%	1.9	0.9	7%	0.9	0.3	−1%
CYP3A4	VC (0.1% DMSO)	1.0	0.3		1.0	0.3		1.0	0.2	
	Rif 10 μM/50 μM	108	9.7	100%	62.3	8.3	100%	68.2	41.3	100%
	Salidroside/1 μM	0.7	0.3	0%	0.9	0.1	0%	0.6	0.1	−1%
	Salidroside/3 μM	0.5	0.2	0%	1.1	0.2	0%	0.5	0.0	−1%
	Salidroside/10 μM	0.5	0.1	0%	1.2	0.4	0%	0.9	0.1	0%
	Salidroside/30 μM	0.4	0.1	−1%	1.0	0.2	0%	0.7	0.2	0%

* Mean of N = 3 replicates.

**Table 6 nutrients-15-03723-t006:** MAO-A and MAO-B (IC_50_) results.

Test Compound	IC50 (µM) *	
	MAO-A	MAO-B
Salidroside (0.5–50 µM)	>50	>50
Positive control	13.8	7.40

Note: MAO Substrate (µM): MAO-A_ kynuramine (20), MAO-B_ kynuramine (20). * Mean of N = 2 replicates.

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
