# Peer review of "In Vitro Evaluation of the Potential for Drug Interactions by Salidroside"

_nutrients, 2023, doi:10.3390/nu15173723_

Round 1
Reviewer 1 Report
Moderate editing of English language required
Author Response
Although the authors did much work, the manuscript could be further improved. Here are some suggestions.
- The manuscript does not clearly explain the differences and advantages between the seabuckthorn studied in this work and the seabuckthornseparated from Rhodiola extract, and why seabuckthorn should be produced by fermentation with genetically engineered Escherichia coli.
Seabuckthorn was not studied in this manuscript. We presented work with Salidroside (produced our E. coli) and demonstrated no effect on CYPs, MAOA/B or OATPs, in contrast to extracts of which are known to contain salidroside. Seabuckthorn is not derived from the Rhodiola rosea.
- The introduction should be concise and avoid unnecessary details. It should also use clear and precise language that is appropriate for the intended audience and purpose of the paper. The contents should be summarised but not just list the detail. e.g., In vitro studies with the different clones, utilizing cDNA-expressed human P45003A4 Supersomes® system indicated an IC50 range of 1.7-3.1 μg/ml,...
Modifications have been made in the Introduction and Discussion to clarify the study. We eliminated the unnecessary detail from the Introduction, as it is summarized in the Discussion along with the contrasting results from our experiments using pure salidroside.
- Most of the references cited are outdated, please refer to the latest research progress. 2004
Reports are from 2004 and later, representing prior studies with Rhodiola rosea extracts or highly cited articles to provide background for Rhodiola rosea use and impact of CYP, MAO or OATP DDI.
- Please provide the structure information of Salidroside. And does salidroside absorbed in its original form? As this study just carried out in vitro studies, so it must be ensured that the salidroside is active in vivo as in its original form.
Salidroside has been widely published and we did not consider adding the structure since this is not a study on biotransformation. The pharmacokinetics in rat suggest that salidroside is orally absorbed intact and has a slow rate of metabolism. While not stated in the manuscript, the oral bioavailability of salidroside in rats was 32%, and over 50% of the salidroside absorbed is recovered unchanged in the urine. The total IV CL was 21 ml/min/kg, so that would mean the metabolic CL is ~10 ml/min/kg, which is low relative to hepatic blood flow of ~70 ml/min/kg (see ref. 9 of manuscript).
- Additionally, the format of the manuscript could be improved.
We hopefully improved the format by reducing the level of detail in the Introduction.
Reviewer 2 Report
The paper describes the DDI-potential of salindroside using in vitro ADME models. The paper is well written and clearly communicates the findings from the study. This paper is eligible for publication after inclusion of the following major comments.
1) The authors in the introduction indicate that the DDI effects of Rhodiola may be due to components other than Salidroside. However, the focus of the paper is DDIs mediated by salidroside which confuses the readers. It is advised to make appropriate modifications to the introduction to avoid this confusion.
2) Please include concentrations of positive control inhibitors and also those of probe substrates for the inhibition and induction experiments.
3) Please include cell viability data for the induction experiments.
4) Why was a different lot used for induction of activity vs. mRNA. It makes sense to evaluate both end points in same hepatocyte lot.
5) In the discussion- CYP Induction was performed for 48h (not 72h) and 30uM salidroside not (30mM)
6) It appears that salidroside has low metabolic stability. Were the salidroside concentrations monitored in the CYP assays? How do you know that salidroside used in the in vitro CYP assays has not disappeared rapidly so as not to cause any DDIs?
Author Response
The paper describes the DDI-potential of salidroside using in vitro ADME models. The paper is well written and clearly communicates the findings from the study. This paper is eligible for publication after inclusion of the following major comments.
1) The authors in the introduction indicate that the DDI effects of Rhodiola may be due to components other than Salidroside. However, the focus of the paper is DDIs mediated by salidroside which confuses the readers. It is advised to make appropriate modifications to the introduction to avoid this confusion.
Modifications have been made in the Introduction and Discussion to clarify the objective of the study to the reader. We eliminated the unnecessary detail from the Introduction, as it is summarized in the Discussion along with the contrasting results from our experiments using pure salidroside.
2) Please include concentrations of positive control inhibitors and also those of probe substrates for the inhibition and induction experiments.
Concentrations have been included.
3) Please include cell viability data for the induction experiments.
Cell viability has been added.
4) Why was a different lot used for induction of activity vs. mRNA. It makes sense to evaluate both end points in same hepatocyte lot.
We agree that the initial lot should have been included, but because we did the preliminary study on enzyme activity and then followed FDA guidelines, using higher concentrations and measuring mRNA levels in three different donor lots. Unfortunately, the initial lot was no longer available.
5) In the discussion- CYP Induction was performed for 48h (not 72h) and 30uM salidroside not (30mM)
Thank you for finding this error. You are correct it should read 48 hours and 30 mM in the Discussion section. We have corrected this.
6) It appears that salidroside has low metabolic stability. Were the salidroside concentrations monitored in the CYP assays? How do you know that salidroside used in the in vitro CYP assays has not disappeared rapidly so as not to cause any DDIs?
Concentrations were not monitored. However, the in vivo pharmacokinetics of salidroside in the rat (reference 9)